# Fusing Sequence and Structural Knowledge by Heterogeneous Models to Accurately and Interpretively Predict Drug–Target Affinity

**DOI:** 10.3390/molecules28248005

**Published:** 2023-12-08

**Authors:** Xin Zeng, Kai-Yang Zhong, Bei Jiang, Yi Li

**Affiliations:** 1College of Mathematics and Computer Science, Dali University, Dali 671003, China; xinzeng@dali.edu.cn (X.Z.); zkaiyang2022@163.com (K.-Y.Z.); 2Yunnan Key Laboratory of Screening and Research on Anti-Pathogenic Plant Resources from Western Yunnan, Dali University, Dali 671000, China; jiangbei@dali.edu.cn

**Keywords:** graph neural network, convolutional neural network, drug–target affinity, sequence and structural knowledge, heterogeneous models

## Abstract

Drug–target affinity (DTA) prediction is crucial for understanding molecular interactions and aiding drug discovery and development. While various computational methods have been proposed for DTA prediction, their predictive accuracy remains limited, failing to delve into the structural nuances of interactions. With increasingly accurate and accessible structure prediction of targets, we developed a novel deep learning model, named S2DTA, to accurately predict DTA by fusing sequence features of drug SMILES, targets, and pockets and their corresponding graph structural features using heterogeneous models based on graph and semantic networks. Experimental findings underscored that complex feature representations imparted negligible enhancements to the model’s performance. However, the integration of heterogeneous models demonstrably bolstered predictive accuracy. In comparison to three state-of-the-art methodologies, such as DeepDTA, GraphDTA, and DeepDTAF, S2DTA’s performance became more evident. It exhibited a 25.2% reduction in mean absolute error (MAE) and a 20.1% decrease in root mean square error (RMSE). Additionally, S2DTA showed some improvements in other crucial metrics, including Pearson Correlation Coefficient (PCC), Spearman, Concordance Index (CI), and *R*^2^, with these metrics experiencing increases of 19.6%, 17.5%, 8.1%, and 49.4%, respectively. Finally, we conducted an interpretability analysis on the effectiveness of S2DTA by bidirectional self-attention mechanism. The analysis results supported that S2DTA was an effective and accurate tool for predicting DTA.

## 1. Introduction

Target proteins are crucial biological macromolecules that play diverse roles in cells. Identifying drug–target interactions is an important process in drug discovery [1]. Calculating a drug molecule (small molecule compound) with high binding affinity against the target is a crucial step in early-stage drug research [2]. The successful identification of drug–target affinity (DTA) has received long-term attention from academia and industry due to its significant role in drug discovery and design [3]. Typically, drug molecules interact with specific targets through direct physicochemical interactions in shallow hydrophobic depressions on the surface of targets or buried deep inside the targets known as pockets [4,5]. These pockets contain charged amino acid residues and hydrophobic residues. Biological experimental observations suggest that the amino acid composition and properties of these pockets are vital determinants for drug–target interactions [4].

Some traditional biological methods to determine DTA are measuring the half maximal inhibitory concentration (IC50), inhibition constant (*K*_i_), and dissociation constant (*K*_d_) between a pair of drugs and targets using bioassays [6]. However, traditional biological experiments to measure DTA are time-consuming, labor-intensive, and resource-intensive [7]. With the increasing number of drug molecules and targets in databases, experimental characterization of all possible drug–target pairs becomes nearly impossible. To address this, numerous computational methods have been developed to predict DTA [8,9,10,11], with some deep learning models based on sequence or/and structural knowledge showing promising results [12,13].

Most DTA prediction models based on deep learning, including DeepDTA [9], DeepDTAF [10], DeepAffinity [14], FusionDTA [15], WideDTA [16], AttentionDTA [17], DeepCDA [18], and MRBDTA [19], primarily rely on drugs’ SMILES (Simplified Molecular Input Line Entry System) [20] and target sequences. These sequence-based methods employed Convolutional Neural Networks (CNNs) [21], Recurrent Neural Networks (RNNs) [22], or the attention mechanism [23] to extract high-level features from the drug SMILES and target sequences. For instance, DeepDTA used three consecutive 1D-convolutional layers to extract local features from drug SMILES and the sequences of targets, which were then passed through three fully connected (FC) layers for DTA prediction. On the other hand, DeepDTAF employed three independent CNN blocks to learn high-level features from the drug SMILES and the sequences of targets and pockets. These CNN blocks consisted of a 1D-convolutional layer for extracting local features from pockets and two 1D dilated convolutional networks [24] for deciphering global features of drugs and targets, respectively. MRBDTA utilized the Trans block with a multi-head attention mechanism to extract molecule features from drug SMILES and the sequences of targets. Then, interaction learning integrated these extracted features for DTA prediction. While sequence-based approaches are effective in predicting DTA, it is evident that their performance can be further enhanced if structural knowledge is incorporated. For example, in DeepDTAF, the pocket knowledge which is essentially obtained from both its sequences and structures may achieve higher performance. Furthermore, these methods are limited in providing a reasonable interpretation of drug–target interactions due to the lack of structural knowledge.

While the structures of targets are challenging to obtain experimentally, some methods, such as GraphDTA [11], MgraphDTA [25], and DGraphDTA [26], have successfully extracted high-level features from a limited number of the structures of drugs and targets to predict DTA. For instance, GraphDTA recognized the structures of small molecules and employed graph convolutional networks (GCNs) [27] to extract graph-based features. In MgraphDTA, multiscale graph neural networks (MGNN) were utilized to extract multiscale features from the input graph of the drug. In DGraphDTA, the structures of targets were represented as graphs based on the contact maps and then combined with the graphs of drugs to predict DTA by two GCNs. Although structure-based methods offer better interpretation, they do have some limitations: (1) the full potential of structural knowledge of drugs and targets remains untapped; (2) most models lack the utilization of semantic knowledge in sequence format, relying solely on structural data for DTA prediction; (3) the 3D spatial positions of the atoms in targets are not fully characterized, as some methods only use contact maps.

In recent years, with the accumulation of experimentally resolved structures of targets and the emergence of algorithms like AlphaFold [28], which can predict the structures of monomeric proteins and proteins without extensive disordered regions, there is a certain degree of relief in obtaining protein structure. To address the problem of insufficient feature extraction from the sequences and structures of drugs and targets in existing methods, a novel deep learning model called S2DTA was proposed. S2DTA predicted DTA by integrating sequence features of drug SMILES, targets, and pockets, and their corresponding graph structural features using a heterogeneous model that combined GCNs and CNNs. Experimental results demonstrated that S2DTA outperformed three other state-of-the-art models by using simple feature representations. Additionally, we conducted an interpretability analysis on the effectiveness of S2DTA using a bidirectional self-attention mechanism, supporting that S2DTA is an effective and accurate tool for predicting DTA.

## 2. Results and Discussion

### 2.1. Limited Performance Improvement by Using Only Sequence Knowledge

Numerous predictive models for DTA primarily rely on the sequences of targets due to the challenge of obtaining the experimental structures of targets. In our study, we conducted comprehensive tests involving both ML and DL models, comparing models that used solely sequence data against a fundamental GCN model that incorporated the structural insights of targets (Table 1). Within the realm of ML models, RF exhibited the most promising performance across all evaluation metrics. Specifically, MAE, RMSE, PCC, Spearman, CI, and *R*^2^ achieved values of 1.221, 1.527, 0.726, 0.713, 0.760, and 0.504, respectively. Turning to DL models, sequence-oriented DL architectures like 1D-CNN and RNN demonstrated relatively weaker performance compared to RF with label encoding. This outcome highlighted that complex DL models did not necessarily lead to performance enhancements. However, 1D-CNN, relying on one-hot encoding, delivered superior performance across all evaluation metrics. This underscored that one-hot encoding provided robust feature representation, enhancing the efficacy of sequence-based DL models. To further enrich the understanding of targets, we incorporated the structural knowledge of targets into the GCN model by characterizing the sequence knowledge of targets as vertex features within the graphs of targets. Empirical findings demonstrated that even a straightforward GCN model with label encoding surpassed RF and 1D-CNN. Notably, when the sequence feature representations of targets employed one-hot encoding, the GCN performed better than RF and 1D-CNN. Specifically, the GCN outperformed RF with a reduction of 0.063 in both MAE and RMSE, and exhibited increments of 0.012, 0.016, 0.009, and 0.04 in PCC, Spearman, CI, and *R*^2^, respectively. All experimental results clearly showed that the simple GCN model with a baseline feature representation outperformed all sequence-based models. Only using the sequence knowledge of targets to predict DTA limited performance improvement.

### 2.2. Performance Comparison of Different Graph-Based Models

Although GraphSAGE has demonstrated superior performance as depicted in Table 1, it is important to note that graph neural networks (GNNs) offer an array of models employing diverse approaches to extract features from graphs. Presently, GNNs are broadly categorized into two types: spectral-based GNNs and spatial-based GNNs. Spectral-based GNNs like GCN and ChebyNet operate on Laplacian matrices, focusing on single graphs. However, for the context of this study, ChebyNet was excluded due to interpretability concerns. Diverging from spectral-based models, spatial-based GNNs such as GraphSAGE, GAT, and GIN gather knowledge from neighboring vertices directly in the spatial domain.

The methodology entailed initial one-hot encoding of drug SMILES and the sequences of targets, utilizing them as the features of vertices in the respective graphs. Subsequently, four distinct graph-based models were harnessed for DTA prediction. The ensuing comparison of experimental results (Table 2) highlighted that GraphSAGE consistently outperformed the other three models across all evaluation metrics. Noteworthiness was its achievement of MAE and RMSE reductions of no less than 0.266 and 0.326 compared to the other models. Moreover, GraphSAGE accomplished improvements of no less than 0.157, 0.156, 0.071, and 0.219 in PCC, Spearman, CI, and *R*^2^, respectively, all of which demonstrated values exceeding 0.5. The better performance of GraphSAGE underscored its efficacy in capturing interactions within the tertiary structures of drugs and targets, reinforcing the significance of structural knowledge in DTA prediction.

### 2.3. Complex Feature Representations Do Not Improve Performance

In addition to employing one-hot encoding, the utilization of complex feature representations for the drug SMILES and target sequences could potentially exert a substantial impact on the performance of DTA prediction. Within the scope of this investigation, a variety of complex feature representations were explored for the sequences of targets, encompassing the physical–chemical properties of amino acids, position-specific scoring matrix (PSSM) [29], Hidden Markov Models matrix (HMM) [30], Protbert [31], CNN, and Word2Vec [32]. For drugs, only the physical–chemical properties of atoms were considered. It is noteworthy that when alternative complex feature representations were applied to the sequences of targets, the drug SMILES representation was employed using one-hot encoding.

Experimental results (Table 3) demonstrated that GraphSAGE attained comparable performance across various representations, including one-hot encoding, Protbert, CNN, and Word2Vec. For instance, the key evaluation metrics like RMSE and *R*^2^ hovered around 1.4 and 0.5, respectively. However, it is pertinent to acknowledge that generating semantic features for the sequences of targets through complex representations like Protbert, CNN, and Word2Vec can be time-intensive. Despite this investment of time, the performance improvements garnered from these complex representations have not been notably substantial.

### 2.4. Impact of Pocket Knowledge on Drug–Target Affinity Prediction

As demonstrated by DeepDTAF, pockets played a crucial role in predicting DTA, and their characteristics directly influenced the binding affinity of drugs with their respective targets. The maximum sequence length for pockets was 125, serving as the fixed sequence length. In cases where the pocket sequences were shorter than 125, padding was applied using zeros. The representation of the sequences of pockets employed one-hot encoding. The structures of pockets were utilized for constructing the graphs, following a methodology consistent with the graph construction of the target.

Three distinct experiments were conducted based on the sequences and structures of targets, pockets, and their combination. Experimental results (Table 4) showed that the performance of GraphSAGE utilizing the sequences and structures of targets was comparable to that using the sequences and structures of pockets across all evaluation metrics. Notably, the MAE, RMSE, PCC, Spearman, CI, and *R*^2^ exhibited values of approximately 1.1, 1.4, 0.7, 0.7, 0.7, and 0.5, respectively. The distribution plots depicting the real and predictive affinity based on targets and pockets were illustrated in Figure 1. However, noteworthy enhancements were observed in the performance of GraphSAGE when incorporating a combination of the sequences and structures of targets and pockets. As compared to using only the knowledge of targets, the MAE and RMSE exhibited reductions of 0.047 (4.2%) and 0.04 (2.8%), respectively. Moreover, improvements of 0.018 (2.4%) in PCC, 0.028 (3.8%) in Spearman, 0.013 (1.7%) in CI, and 0.024 (4.2%) in *R*^2^ were realized. Similarly, in comparison to using only the knowledge of pockets, the MAE and RMSE experienced decreases of 0.021 (1.9%) and 0.43 (3%), while enhancements of 0.017 (2.2%) in PCC, 0.018 (2.4%) in Spearman, 0.009 (1.2%) in CI, and 0.026 (4.6%) in *R*^2^ were achieved. The distribution of real and predictive affinity, incorporating both the sequence and structural knowledge of targets and pockets, is visualized in Figure 1.

### 2.5. Fusing Sequence and Structure Knowledge with Heterogeneous Models

Although the semantic features extracted from the sequences of targets using Protbert, CNN, and Word2Vec did not lead to an improvement in the performance of GraphSAGE, fusing sequence knowledge with CNN to create an end-to-end model could enhance the performance. The process involved employing one-hot encoding for the sequences of targets, which was then fed into a CNN to extract sequence knowledge. This extracted sequence knowledge was utilized as vertex features within the graphs of targets, subsequently combined with the graphs of targets and input into a three-layer GCN network to extract the fusion knowledge. This approach was also used to extract the sequence features of pockets, but not to drug SMILES. Finally, the knowledge extracted from drugs, targets, and pockets by three independent three-layer GCN networks was concatenated and fed into the FC network to predict DTA.

Experimental results (Table 5) demonstrated some enhancements in performance by integrating CNN into the model alongside GCN and FC. In comparison to the model without CNN (Table 4), the MAE and RMSE were reduced by 0.090 (8.3%) and 0.078 (5.7%), respectively. Moreover, the PCC, Spearman, CI, and *R*^2^ exhibited improvements of 0.025 (3.2%), 0.027 (3.5%), 0.013 (1.6%) and 0.044 (7.4%), respectively. In addition to CNN, several other sequence-processing models, such as Long Short-Term Memory (LSTM) [33] and Bidirectional LSTM (BiLSTM) [34], are also capable of extracting sequence knowledge. Extensive experiments were conducted on these commonly utilized sequence-processing models, with the findings (Table 5) indicating that integrating sequence knowledge with CNN consistently yielded superior performance. The distribution of predictive affinity resulting from the fusion of sequence knowledge with sequence-processing models is depicted in Figure 2.

In summary, the proposed method, denoted as S2DTA, which encompassed CNN, GCN, and FC, exhibited the highest performance by amalgamating sequence and structural knowledge from drugs, targets, and pockets. The process involved extracting the sequence knowledge from the sequences of targets and pockets, and incorporating structures and the extracted sequence knowledge from targets and pockets into GCN to attain the high-level fused features. For drugs, high-level features were directly extracted from drug SMILES and their structures through GCN. Ultimately, all of these features were concatenated and fed into a FC network to predict DTA.

### 2.6. Comparison with State-of-the-Art Methods

To comprehensively assess the predictive capability of S2DTA for DTA prediction, we conducted a comparative analysis against three deep learning methods: DeepDTA, DeepDTAF, and GraphDTA. Firstly, we trained the three compared methods on the training set provided in this study. Then employing the test dataset sourced from the PDBbind database (2016 version), S2DTA emerged as the superior performer across all evaluation metrics (Table 6). Experimental findings showed that S2DTA exhibited a 25.2% reduction in MAE and a 20.1% decrease in RMSE when compared to the three state-of-the-art methodologies. Additionally, S2DTA displayed some improvements in other vital metrics: PCC, Spearman, CI, and *R*^2^, with increases of 19.6%, 17.5%, 8.1%, and 49.4%, respectively. Finally, these experiments underscored the robustness and reliability of S2DTA as a predictive model for DTA.

### 2.7. Performance Comparison between the Real Structures and Predictive Structures of Targets

Currently, a lot of targets possess known sequences but lack clear structures. This limitation hampers the application of S2DTA. However, the emergence of AlphaFold offers the potential to predict the structures of monomeric proteins and proteins without extensive disordered regions. To evaluate the adaptability of S2DTA on these predictive structures of targets, we selected 225 sequences from the test dataset, each with a length of approximately 500 amino acid residues. Employing AlphaFold, we obtained predictive structures for these 225 sequences. The outcomes of S2DTA using both real and predictive structures of targets are presented in Table 7. For S2DTA utilizing the predictive structures of targets, the MAE and RMSE stood at 1.133 (compared to 1.046 with real structures) and 1.432 (1.316), respectively. The PCC, Spearman, CI, and *R*^2^ for S2DTA employing the predictive structures of targets were 0.748 (compared to 0.792 with real structures), 0.749 (0.787), 0.78 (0.799), and 0.558 (0.627), respectively. The evaluation metrics collectively indicated that the performance of S2DTA was similar to that when using real structures of targets. S2DTA demonstrated good generalization proficiency when using the predictive structures of targets.

### 2.8. Structure-Based Interpretable Analysis of S2DTA

To explain the effectiveness of S2DTA, we employed a bidirectional self-attention mechanism to extract crucial information regarding drug–target binding affinity. This was performed after the pooling operation in S2DTA, resulting in attention score matrices that capture the interactions between drugs and targets. By making use of the structural knowledge of drugs and targets, we aimed to provide interpretability for the effectiveness of S2DTA. The pockets are the primary sites where drug–target interactions take place. Hence, we conducted a structural explanatory analysis of S2DTA by explaining the atoms constituting the drugs and the amino acids present in the targets, using different attention score matrices. For drugs, we computed the binding data between each atom (excluding hydrogen atoms) and all amino acids in the targets. Subsequently, we performed data statistics (Table 8) to determine whether the top 5, top 10, and top 20 amino acids in the attention score matrices were located within the pockets. The statistical findings revealed that the top 5, top 10, and top 20 amino acids within the pockets accounted for 13.7%, 12.8%, and 12.4%, respectively. In the upper row of Figure 3, we displayed four targets for drug binding. In the middle row of Figure 3, we showcased the top 10 amino acids (depicted in purple) identified by four drugs (PubChem ID: 43158872, 444128, 44629533, and 53308636) within the pockets of their respective targets (PDB ID: 3AO4, 1A30, 2QNQ, and 3O9I). These proportions were relatively low due to the length of target sequences, which made it challenging to identify any interactions between each atom and the amino acids in the pockets. However, S2DTA demonstrated its ability to effectively identify the amino acids within the pockets that interacted with the atoms. Conversely, we also examined the interaction between each amino acid and all atoms (excluding hydrogen atoms). Statistical analysis (Table 8) was performed on the top 3, top 5, and top 8 atoms in the attention score matrices. Interestingly, these atoms accounted for 48.6%, 56%, and 64.8% of the total number of atoms in the drugs, respectively. The lower row of Figure 3 presented the visualization results of the top 5 atoms (represented by the purple logo) with the highest scores in the interactions between the amino acids constituting the targets and the corresponding drugs. These above statistical results indicated that S2DTA could effectively identify the specific atoms that interacted with the amino acids. Hence, the attention score matrices derived from bidirectional self-attention served as a key explanation for the effectiveness of S2DTA in accurately predicting drug–target binding affinity. This underscores the reliability and interpretability of S2DTA as a practical tool for drug–target affinity prediction.

### 2.9. Case Study

To further validate the effectiveness of the model, we compared S2DTA with DeepDTAF and MM-GBSA/PBSA [35] in specific samples. Firstly, we randomly selected 50 drug–target binding samples from the test set of the PDBbind database (2016 version) and calculated the affinity values of all samples using S2DTA, the predictive model provided by the advanced method DeepDTAF, and MM-GBSA/PBSA to calculate the binding free energy of each sample. We then normalized the absolute values of the real affinity values, predictive affinity values, and binding free energy (most of which are negative). Finally, we evaluated the correlation between the normalized real affinity values and the normalized predictive values using PCC and *R*^2^. Our experimental results (Figure 4) showed that the binding energy calculated by MM-GBSA/PBSA has a poor linear fit with the real affinity values and two evaluation metrics (PCC = 0.597, *R*^2^ = 0.356) reflected their weak correlation. However, both S2DTA (PCC = 0.969, *R*^2^ = 0.941) and DeepDTAF (PCC = 0.921, *R*^2^ = 0.847) exhibited strong correlations with the real affinity values in terms of linear fit and evaluation metrics. Furthermore, we found that S2DTA demonstrated slightly better performance than DeepDTAF, as confirmed by the linear fitting degree and the two evaluation metrics PCC and *R*^2^ in Figure 4. Therefore, our experimental results suggest that S2DTA is an effective and reliable tool for predicting drug–target binding affinity. 

## 3. Materials and Methods

### 3.1. Datasets

In line with current competing methods [9,10,11], we utilized a benchmark dataset from the PDBbind database (2016 version) [36] to train, validate, and test our model. This dataset comprises 13,283 high-resolution structures of drug–target complexes sourced from the Protein Data Bank (PDB, https://www.rcsb.org (accessed on 16 September 2023)), along with experimental values of DTA, typically represented by $pK_{d}$. The training and testing datasets were constructed from the PDBbind database (2016 version), resulting in 12,993 and 290 samples, respectively. To ensure the training efficiency of the model, the sequence length of targets was fixed at 2100 amino acids to cover 98.74% of samples (Appendix A) of the training dataset. A total of four samples were excluded due to processing issues with the Biopython package [37]. Consequently, the training dataset comprises 12,823 samples, while the testing dataset has 289 samples. The maximum length of drug SMILES was chosen as the fixed length. Sequences of targets or drug SMILES shorter than their fixed lengths were zero-padded.

### 3.2. Sequence Representation

We utilized label encoding and one-hot encoding as baseline representations to depict the 20 commonly used amino acids in targets and the 27 different atoms in drugs. Label encoding employs a single number to differentiate objects, while one-hot encoding maps encoded objects to status registers, generating encoding vectors. To characterize the diverse amino acids of targets and atoms of drugs, we selected 24 (Appendix A) [38] and 9 physical–chemical properties (Appendix A), respectively. For the sequence features of targets, we extracted the PSSM and utilized HHBlits for aligning homologous sequences to achieve the HMM. PSSM and HMM matrices, with dimensions of 20, encapsulate the features of the 20 commonly used amino acids in the sequences of targets. Furthermore, we harnessed multiple sequence-processing models, including Protbert, CNN, and Word2Vec, to extract semantic features from the sequences of targets.

### 3.3. Structure Representation

To exploit structural insight, we devised a graph-based data format for drugs, targets, and pockets. In the case of drugs, the individual atoms composing the drug formed the vertices of its molecular graph. Covalent bonds from the .sdf or .mol2 files of drugs were employed to establish edges. Regarding the graphs of targets, the carbon α atoms of amino acid residues in the structures of targets were designated as vertices, and edges were established between any two carbon α atoms within a Euclidean distance of 8 Å. The structure representation of the pocket was consistent with the target.

### 3.4. Proposed S2DTA Approach

In this study, a deep learning method consisting of heterogeneous models, called S2DTA (see Figure 5), was proposed to accurately and interpretively predict DTA by fusing the sequence and structural knowledge of drugs and targets. S2DTA is an end-to-end model that contained four modules: data input module, sequence learning module, structure learning module, and feature fusion module.

(1) Data input module: S2DTA processed the drug SMILES and their structure data, targets, and pockets, where input sequences were represented as a one-hot encoded feature vector, and structure data included the molecular graphs for drugs and the graphs of targets and pockets.

(2) Sequence learning module: A 1D-CNN layer was employed to extract semantic features from both targets and pockets based on their sequence data. The 20-dimensional sequence data of targets underwent convolution with a kernel size of 5, and a stride of 1, resulting in an output dimension of 128. Meanwhile, the 20-dimensional sequence data of pockets utilized a convolution kernel of size 3, with the remaining parameters identical to those used for targets.

(3) Structure learning module: Three independent GCN networks, utilizing the message-passing mechanism, were employed to extract high-level features from vertices within the graph-based representation of drugs, targets, and pockets. Each GCN network consisted of three layers of graph convolutions. In the case of targets and pockets, the feature dimension of each vertex was 128, with the three graph convolutional layers having dimensions of 32, 64, and 128, sequentially. A 128-dimensional feature vector obtained from the final graph convolutional layer for targets and pockets was linearly transformed into a 256-dimensional feature vector. For drugs, the 27-dimensional sequence data consisted in the feature dimension of their vertices, and the dimensions of the three graph convolutional layers were 32, 64, and 128. Similarly, a 128-dimensional feature vector of the drug from the third graph convolutional layer was linearly transformed into a 256-dimensional feature vector.

(4) Feature fusion module: The features extracted from drugs, targets, and pockets through GCN networks were concatenated, resulting in a comprehensive 256-dimensional feature vector. This feature vector was then fed into a FC network to predict DTA. The FC network comprised two linear layers. The initial linear layer, acting as a hidden layer, comprised 256 neurons. The subsequent linear layer, serving as the output layer, consisted of a single neuron responsible for producing the affinity.

### 3.5. Training Process

S2DTA underwent a training process spanning 300 iterations, with optimization carried out using the Adam optimizer [39]. The LeakyReLU activation function [40] was employed, and mean square error (MSE) served as the chosen loss function. The hyperparameters adopted for S2DTA were as follows: a learning rate of 0.0001, a batch size of 128, and a dropout rate of 0.4.

### 3.6. Comparison Strategy

A comprehensive set of machine learning (ML) and deep learning (DL) models, and MM-GBSA/PBSA were employed in this study to establish a robust baseline for evaluation. The ML models encompassed ridge regression (RR), K nearest neighbor (KNN), support vector machine regression (SVR), decision tree (DT), and random forest (RF). On the other hand, the DL models comprised 1D-CNN and RNN. Additionally, a variety of graph-based models, including GCN, ChebyNet [41], GraphSAGE [42], GAT [43], and GIN [44], were tested.

The implementation of ML models was facilitated through the scikit-learn library (https://scikit-learn.org (accessed on 16 September 2023)), while the DL models were created using the Pytorch package (https://pytorch.org/ (accessed on 16 September 2023)). Furthermore, the graph-based models applied the capabilities of the Deep Graph Library (DGL, https://www.dgl.ai/ (accessed on 16 September 2023)) to carry out their computations and operations.

### 3.7. Evaluation Metrics

Six evaluation metrics including mean absolute error (MAE), root mean square error (RMSE), Pearson Correlation Coefficient (PCC), Spearman, Concordance Index (CI), and *R*-squared (*R*^2^) were used to comprehensively evaluate the performance of the model. MAE is used to measure the mean absolute error between predictive and real values. It reflects the size of the real prediction error; RMSE is often used to measure the deviation between predictive and real values; PCC is used to measure the mutual relationship between two variables; Spearman is a nonparametric measure of the dependence of two variables; CI is an index used to evaluate the prediction accuracy of the model; *R*^2^ is mainly used to measure how well the predictive value fits the real value.

## 4. Conclusions

In this study, we proposed S2DTA, a graph-based deep learning model designed to predict DTA with accuracy and interpretability. By leveraging heterogeneous models and fusing sequence and structural knowledge, S2DTA demonstrated novel advancements in DTA prediction. The S2DTA model introduces two main innovations. Firstly, it integrates sequence features of drug SMILES, targets, and pockets and their corresponding structural features. This builds upon previous research which primarily focused on drug SMILES and the sequences of targets, thereby enhancing the overall performance of the model. Secondly, the incorporation of structural knowledge not only boosts predictive accuracy but also offers interpretability for DTA. However, it is important to recognize three limitations of S2DTA.

AlphaFold can only effectively predict the structures of monomeric proteins and proteins without extensive disordered regions.

In cases where pocket knowledge for drug–target binding is missing from datasets, precise pocket identification tools or biological experiments become necessary to provide this crucial knowledge.

S2DTA is a deep-learning model specifically designed to predict drug–target binding affinity. It should be noted that there are certain limitations when it comes to predicting the binding affinity between drugs and other entities, such as drugs and nucleic acids or drugs and other drugs.

While S2DTA has demonstrated itself as an effective and interpretable predictive model for DTA, its refinement is still underway with a focus on two key aspects: (1) Although S2DTA utilizes CNN effectively to extract local sequence knowledge, it will be advanced to incorporate the extraction of long-range dependence knowledge. This expansion will broaden the understanding of the intricate relationship of the model within sequences and contribute to its predictive performance. (2) Recognizing the substantial impact that surface pocket features have on DTA, S2DTA will undergo further development to extract and integrate these features into its predictive framework.

In summary, the accessibility of drug molecular maps and target tertiary structures has paved the way for exciting prospects in drug–target affinity research. Applying CNN, GNN, and 3D-CNN models to extract high-level features from the drug SMILES, target sequences, and their structures, along with complementary fusion of the extracted heterogeneous knowledge, enables us to fully harness the potential of both sequence and structural data. Undoubtedly, the integration of deep learning with the drug SMILES, target sequences, and their structures will emerge as a prominent and promising research direction in the future.

## Figures and Tables

**Figure 1 molecules-28-08005-f001:**
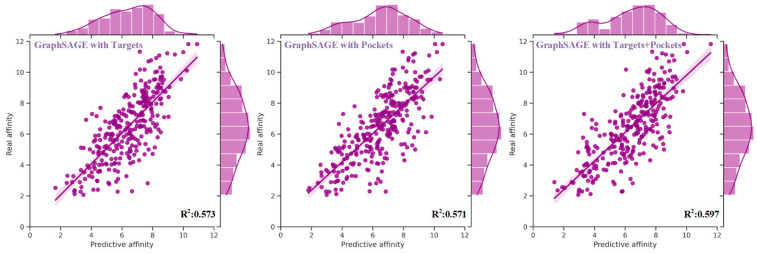
The distribution of real and predictive affinity of GraphSAGE with targets, pockets, and their combination based on the test set. The performance of GraphSAGE, when harnessed in conjunction with both the sequence and structural knowledge of targets and pockets, surpassed that achieved by solely relying on the sequence and structural features of targets or pockets individually. Specifically, as compared to using only the features of targets, the MAE and RMSE exhibited reductions of 0.047 and 0.04, respectively. Moreover, improvements of 0.018 in PCC, 0.028 in Spearman, 0.013 in CI, and 0.024 in *R*^2^ were realized. Similarly, in comparison to using only the features of pockets, the MAE and RMSE experienced decreases of 0.021 and 0.43, while enhancements of 0.017 in PCC, 0.018 in Spearman, 0.009 in CI, and 0.026 in *R*^2^ were achieved.

**Figure 2 molecules-28-08005-f002:**
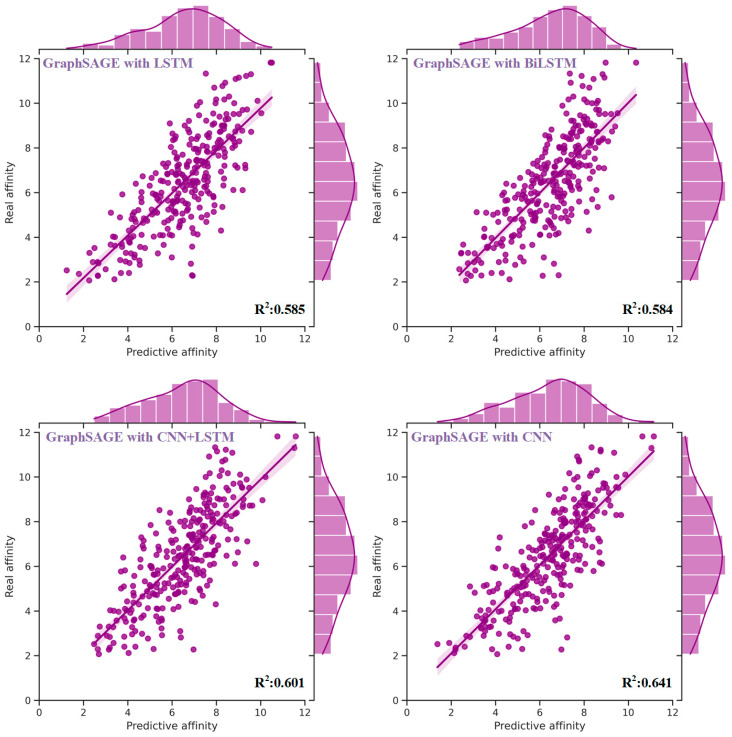
The distribution of real and predictive affinity of heterogeneous models to fuse sequence and structural knowledge from drugs, targets, and pockets based on the test set. Among multiple sequence-processing models, including CNN, LSTM, BiLSTM, and their combinations, the end-to-end heterogeneous model composed of CNN and GCN exhibited the best performance. In comparison to the model without CNN, the MAE and RMSE were reduced by 8.3% and 5.7%, respectively. Moreover, the PCC, Spearman, CI, and *R*^2^ exhibited improvements of 3.2%, 3.5%, 1.6% and 7.4%, respectively.

**Figure 3 molecules-28-08005-f003:**
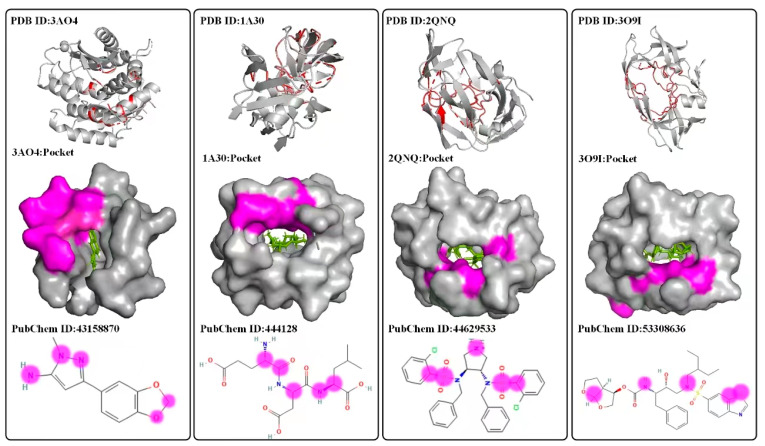
The visualization results of drug–target binding affinity by a bidirectional self-attention mechanism for explaining the effectiveness of S2DTA. In the **upper row**, we present the targets of amino acids with higher drug interaction scores. In the **middle row**, we show the top 10 amino acids (depicted in purple) identified by four drugs (PubChem ID: 43158872, 444128, 44629533, and 53308636) within the pockets of their respective targets (PDBID: 3AO4, 1A30, 2QNQ, and 3O9I). The statistical findings reveal that the top 5, top 10, and top 20 amino acids within the pockets account for 13.7%, 12.8%, and 12.4%, respectively. The **lower row** presents the visualization results of the top 5 atoms (represented by the purple logo) with the highest scores in the interactions between the amino acids constituting the targets and the corresponding drugs. Statistical analysis was performed on the top 3, top 5, and top 8 atoms in the attention score matrices. These atoms accounted for 48.6%, 56%, and 64.8% of the total number of atoms in the drugs, respectively.

**Figure 4 molecules-28-08005-f004:**
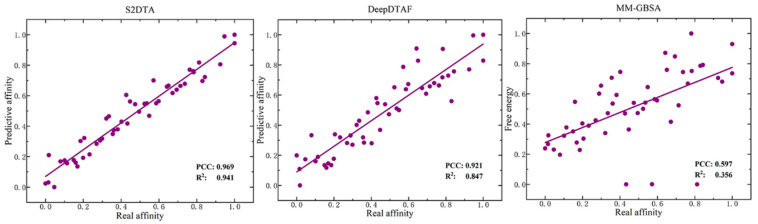
The correlation analysis between the real affinity values and predictive values of S2DTA, DeepDTAF, and MM-GBSA/PBSA. The results show that the real affinity values of the selected samples exhibited a strong linear relationship with the predictive affinity values of both S2DTA (PCC = 0.969, *R*^2^ = 0.941) and DeepDTAF (PCC = 0.921, *R*^2^ = 0.847). Notably, S2DTA showed slightly superior performance compared to DeepDTAF. In contrast, we observed that the real affinity values of the selected samples did not align well with the free energy predicted by MM-GBSA/PBSA (PCC = 0.597, *R*^2^ = 0.356) in terms of a linear relationship, and its predictive values exhibited a weak correlation with the real affinity values.

**Figure 5 molecules-28-08005-f005:**
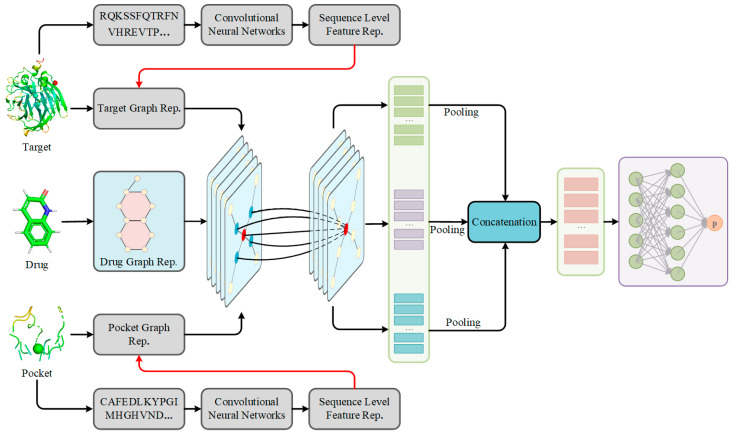
The architecture of S2DTA. S2DTA is an advanced model designed to predict drug–target affinity (DTA) by leveraging sequence features of drug SMILES, targets, and pockets, and their corresponding structural features. This model comprises four essential modules: (1) The data input module was responsible for representing the sequence and structural data of drugs, targets, and pockets. (2) Within the sequence learning module, a 1D-CNN layer was employed to extract semantic features from both targets and pockets, based on their sequence data. (3) In the structure learning module, three independent Graph Convolutional Networks (GCNs) were employed to extract high-level features from the vertices present in the graph-based representation of drugs, targets, and pockets. (4) The feature fusion module encompasses a two-layer fully connected (FC) network, which played a pivotal role in predicting DTA by integrating the extracted features.

**Table 1 molecules-28-08005-t001:** Performance of machine learning and deep learning models using only sequence knowledge compared to graph-based models fusing structure knowledge.

Category	Model	Feature Representation	MAE	RMSE	PCC	Spearman	CI	*R* ^2^
Machine learning models	RR	label encoding	1.471	1.864	0.515	0.521	0.685	0.261
	KNN	label encoding	1.377	1.732	0.604	0.565	0.703	0.362
	DT	label encoding	1.409	1.882	0.590	0.577	0.707	0.247
	SVR	label encoding	1.749	2.150	0.156	0.231	0.527	0.017
	RF	label encoding	1.221	1.527	0.726	0.713	0.760	0.504
Sequence-based deep learning models	1D-CNN	label encoding	1.569	1.938	0.535	0.531	0.683	0.201
		one-hot encoding	1.159	1.473	0.752	0.741	0.776	0.539
	RNN	label encoding	1.815	2.227	0.175	0.329	0.611	−0.054
		one-hot encoding	1.669	2.045	0.459	0.460	0.657	0.111
Structure-based deep learning models	GraphSAGE	label encoding	1.288	1.599	0.675	0.664	0.739	0.449
		one-hot encoding	1.127	1.417	0.758	0.745	0.778	0.573

**Table 2 molecules-28-08005-t002:** Performance comparison of different graph-based models.

Model	Feature Representation	MAE	RMSE	PCC	Spearman	CI	*R* ^2^
GCN	one-hot encoding	1.393	1.743	0.601	0.589	0.707	0.354
GIN	one-hot encoding	1.524	1.842	0.532	0.549	0.692	0.279
GAT	one-hot encoding	1.708	2.091	0.296	0.313	0.606	0.070
GraphSAGE	one-hot encoding	1.127	1.417	0.758	0.745	0.778	0.573

**Table 3 molecules-28-08005-t003:** Performance comparison of GraphSAGE with different feature representations.

Feature Representation	MAE	RMSE	PCC	Spearman	CI	*R* ^2^
one-hot encoding	1.127	1.417	0.758	0.745	0.778	0.573
physical–chemical properties	1.444	1.770	0.583	0.585	0.706	0.334
PSSM	1.236	1.526	0.722	0.713	0.761	0.505
HMM	1.342	1.664	0.644	0.639	0.729	0.411
semantic feature (Protbert)	1.139	1.485	0.734	0.720	0.768	0.531
semantic feature (CNN)	1.158	1.477	0.733	0.723	0.769	0.536
semantic feature (Word2Vec)	1.136	1.411	0.760	0.746	0.777	0.577

**Table 4 molecules-28-08005-t004:** Performance comparison of GraphSAGE with targets or combining the knowledge of pockets.

Name	MAE	RMSE	PCC	Spearman	CI	*R* ^2^
Targets	1.127	1.417	0.758	0.745	0.778	0.573
Pockets	1.101	1.420	0.759	0.755	0.782	0.571
Targets + Pockets	1.080	1.377	0.776	0.773	0.791	0.597

**Table 5 molecules-28-08005-t005:** Performance comparison of heterogeneous models with fused sequence and structural knowledge.

Sequence Processing Model	Structural Model	MAE	RMSE	PCC	Spearman	CI	*R* ^2^
LSTM	GraphSAGE	1.084	1.397	0.766	0.749	0.781	0.585
BiLSTM		1.100	1.399	0.764	0.765	0.785	0.584
CNN + LSTM		1.075	1.370	0.776	0.771	0.789	0.601
CNN		0.990	1.299	0.801	0.800	0.804	0.641

**Table 6 molecules-28-08005-t006:** Performance comparison of S2DTA with three state-of-the-art methods.

Model	MAE	RMSE	PCC	Spearman	CI	*R* ^2^
DeepDTA	1.323	1.641	0.667	0.681	0.744	0.428
GraphDTA	1.367	1.715	0.618	0.612	0.717	0.374
DeepDTAF	1.327	1.625	0.670	0.654	0.734	0.429
S2DTA	0.990	1.299	0.801	0.800	0.804	0.641

**Table 7 molecules-28-08005-t007:** Performance comparison of S2DTA based on the real and predictive structures of targets.

Model	Structures of Targets	MAE	RMSE	PCC	Spearman	CI	*R* ^2^
S2DTA	Real structures	1.046	1.316	0.792	0.787	0.799	0.627
	Predictive structures	1.133	1.432	0.748	0.749	0.780	0.558

**Table 8 molecules-28-08005-t008:** Data statistics for interpretability analysis of S2DTA.

Atoms of Drugs			Amino Acids of Targets		
Top 5	Top 10	Top 20	Top 3	Top 5	Top 8
13.7%	12.8%	12.4%	48.6%	56%	64.8%

## Data Availability

The source data and code repository can be accessed at https://github.com/dldxzx/S2DTA (accessed on 16 September 2023).

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
