# Peer review of "Fusing Sequence and Structural Knowledge by Heterogeneous Models to Accurately and Interpretively Predict Drug–Target Affinity"

_molecules, 2023, doi:10.3390/molecules28248005_

Round 1
Reviewer 1 Report
Comments and Suggestions for Authors
The authors have presented a naturally built-up model, with increasing complexity of the AI model. The final model was compared to other state-of-the-art methods from literature and outperformed for all statistical quantities. Its performance is even very promising in case of target structures predicted by e.g. AlphaFold.
Some minor questions/suggestions:
1) Maybe the authors could indicate which of the statistical quantities they deem more relevant to consider when assessing the model quality.
2) The attention scores in the interpretability analysis do not uniformly identify the pocket as the region of interaction. Does this mean that better interactions are to be expected outside the pockets?
3) In the SI, the caption of Table S1 needs to mention all quantities by name and unit.
Comments on the Quality of English LanguageGood quality of English with some minor spelling errors.
Reviewer 2 Report
Comments and Suggestions for Authors
Before endorsing this work, I have a few major comments that should be addressed:
1) As explained in section 2.3, why a Euclidean distance of 8Å was chosen. Can authors show the impact on edges by changing this distance to 7Å or even to 6Å?
2) DFA is predicted, which is great, but authors are suggested to present a case-study as an example of at-least one target-drug interaction and compare the DFA with their proposed method versus the standard MM-GBSA/PBSA values, and/or other affinity scores standard in literature.
3) Can this method be translated to Nucleic acids? I do not see any discussion on this point. Please clarify this.
Reviewer 3 Report
Comments and Suggestions for Authors
GENERAL
This manuscript describes a computational approach (S2DTA) for predicting drug-target affinity (DTA) based on models that combine sequence and structural data on targets with structural and physical-chemical properties of drugs. Performance of the approach was assessed by metrics including RMSE, MAE, PCC, SCC, CI and R^2. The following general issues should be satisfactorily addressed:
1. Writing and presentation. The presentation was fairly clear. However, at times the writing style bordered on hyperbole or a chatbot product, resembling a commercial rather than a scientific paper. These distracting features could be addessed by removing some of the superlatives and letting the results speak for themselves.
2. Dataset partitioning. It appears from the description in the Methods that from the PDBbind database, the training set comprised 12883 and the test set included 290 samples. If this is so, the test set was only 2.25% of the total. Why was the partitioning so heavily weighted toward the training set? Furthermore, was the training set subdivided into training and validation subsets? If so, please indicate the number of samples in each subset.
3. Realm of applicability. The authors need to specify the chemical space of the drugs (or drug candidates) covered by the chosen samples as well as the types of receptors or enzymes that were chosen. For example, what were the relative contributions from, e.g., membrane-bound receptors, protein kinases, nuclear receptors, P450s, esterases, or other receptor/enzyme classes?
4. Model evaluation. One of the metrics was the Pearson correlation coefficient. To apply this, the data should follow a Gaussian distribution -- was this the case? In addition, why was the Mathhews correlation coefficient not used?
5. Model comparison. It would have been of interest to have compared S2DTA with models used by commercial software, such as Simulations Plus ADMET Predictor/Modeler.
6. Overtraining. What steps were taken to avoid overtaining of models?
Specific issues are listed below by line numbers.
SPECIFIC
1. Line 15. "...sequence and structural knowledge of drugs, targets, and pockets...." This implies that drugs have sequences. In addition, whereas the sequence of a pocket could be indicated, it would be out of context of the entire target sequence. It would be good to make explicit which attribute(s) apply to each item (drugs, targets, and pockets).
2. Line 18-19. Which three state of the art methodologies? These should be identified.
3. Line 19-26. Some examples of distracting words: "...supremacy ... strikingly ...showcased ...impressive... pivotal ... remarkable ... remarkable ... fully proving ... valuable.
4. Line 37-39. The binding sites of drugs are not limited to "... shallow hydrophobic depressions on the surface of targets ...." They can be buried deep within proteins and can contain charged amino acid residues as well as hydrophobic residues.
5. Line 42-43. IC50 values are not necessarily the most accurate experimental measures of DTAs. In many cases, it is better to use kinetic (rate) constants or equilibrium constants. Futhermore, in cases of covalent binding, kinetic constants must be used.
6. Line 52. Here again, there can be some confusion about modifiers (" ... sequence knowledge of drugs and targets.")
7. Line 85-86. This sentence is awkward and requires rewriting.
8. Line 86-87. This statement is generally true, but currently only for monomeric proteins and proteins without extensive disordered regions. Moreover, AlphaFold does not currently include information about association with membranes or the inclusion of cofactors, ligands, or conserved water molecules that can participate in ligand binding.
9. Line 94. "...fully proving ...." This is an overstatement. It should be softened at least somewhat. The results support rather than fully prove the accuracy of S2DTA. In addition, "value" is a subjective term that would be difficult to prove objectively.
10. Line 100. Define "high quality". Additionally, indicate if the complexes including only actual "drugs" or if the samples included just compounds that have not yet been approved as drugs.
11. Line 115-117. "physical-chemical properties" is repeated.
12. Line 190-191. Define "meticulously tested".
13. Line 193. Define "meticulously crafted".
14. Line 198-200. Were the data normally distributed? For Pearson at least, this would be a requirement. Why was the Mathhews correlation coefficient not used?
15. Line 218. "didn't". Contractions should be avoided in formal, professional, or scientific writing.
16. Line 227. "outshined". This is a distracting term that seems too colloquial and subjective.
17. Figure 2. More explanation is needed to describe the plots. In addition, it would be helpful to color-code the data markers according to test and training sets. Finally, it would be good to indicate the R^2 value on each plot.
18. Line 320. The construction "...it would not be used to drug SMILES" requires rewriting.
19. Figure 3. See comments for Figure 2.
20. Lines 350-361. Again, some distracting terms are employed, e.g., "...superiority .. showcasing ... impressive ... vividly ..."
21. Line 365. What constitutes a "signficiant" number? Use caution with terms that imply statistical significance.
22. Line 386 and throughout the manuscript, the term "leveraging" seems overused.
23. Line 391-392. If hydrogen atoms were excluded from the binding data, this would exclude a very important contributor, namely, hydrogen-bonding.
24. Figure 4. More explanation is needed in the legend to this figure. In addition, these examples all appear to be relatively simple globular proteins. Are these typical of all of the samples selected from the PDBbind database.
25. Line 434. "it's". Avoid contractions.
26. Line 450-451. "Leveraging the power" is a rather loaded phrase. What does "power" mean in this context?
END OF COMMENTS
Comments on the Quality of English Language
Please see the general and specific comments above that also address comments on English language usage.
Round 2
Reviewer 2 Report
Comments and Suggestions for Authors
Acceptable!
Author Response
No review comments.
Reviewer 3 Report
Comments and Suggestions for Authors
The authors have provided a satisfactory point-by-point description of their responses to the comments, and the revised manuscript now appears to be publishable.
Author Response
No review comments